# An Efficient and Robust Method for Chest X-ray Rib Suppression That Improves Pulmonary Abnormality Diagnosis

**DOI:** 10.3390/diagnostics13091652

**Published:** 2023-05-08

**Authors:** Di Xu, Qifan Xu, Kevin Nhieu, Dan Ruan, Ke Sheng

**Affiliations:** 1Department of Radiation Oncology, University of California at Los Angeles, Los Angeles, CA 90095, USA; dixu@mednet.ucla.edu (D.X.); qifanxu@mednet.ucla.edu (Q.X.); kknhien7@g.ucla.edu (K.N.); druan@mednet.ucla.edu (D.R.); 2Department of Radiation Oncology, University of California at San Francisco, San Francisco, CA 94115, USA

**Keywords:** chest X-rays, deep learning, rib suppression, computer aided diagnosis

## Abstract

Background: Suppression of thoracic bone shadows on chest X-rays (CXRs) can improve the diagnosis of pulmonary disease. Previous approaches can be categorized as either unsupervised physical models or supervised deep learning models. Physical models can remove the entire ribcage and preserve the morphological lung details but are impractical due to the extremely long processing time. Machine learning (ML) methods are computationally efficient but are limited by the available ground truth (GT) for effective and robust training, resulting in suboptimal results. Purpose: To improve bone shadow suppression, we propose a generalizable yet efficient workflow for CXR rib suppression by combining physical and ML methods. Materials and Method: Our pipeline consists of two stages: (1) pair generation with GT bone shadows eliminated by a physical model in spatially transformed gradient fields; and (2) a fully supervised image denoising network trained on stage-one datasets for fast rib removal from incoming CXRs. For stage two, we designed a densely connected network called SADXNet, combined with a peak signal-to-noise ratio and a multi-scale structure similarity index measure as the loss function to suppress the bony structures. SADXNet organizes the spatial filters in a U shape and preserves the feature map dimension throughout the network flow. Results: Visually, SADXNet can suppress the rib edges near the lung wall/vertebra without compromising the vessel/abnormality conspicuity. Quantitively, it achieves an RMSE of ~0 compared with the physical model generated GTs, during testing with one prediction in <1 s. Downstream tasks, including lung nodule detection as well as common lung disease classification and localization, are used to provide task-specific evaluations of our rib suppression mechanism. We observed a 3.23% and 6.62% AUC increase, as well as 203 (1273 to 1070) and 385 (3029 to 2644) absolute false positive decreases for lung nodule detection and common lung disease localization, respectively. Conclusion: Through learning from image pairs generated from the physical model, the proposed SADXNet can make a robust sub-second prediction without losing fidelity. Quantitative outcomes from downstream validation further underpin the superiority of SADXNet and the training ML-based rib suppression approaches from the physical model yielded dataset. The training images and SADXNet are provided in the manuscript.

## 1. Introduction

Respiratory diseases are among the major causes of morbidity and mortality globally, and the prevalence of pulmonary diseases has steadily increased [1,2]. Timely diagnosis is critical for effective intervention. Among all imaging tools, a chest X-ray (CXR) is the most widely used for pre-screening thoracic anomalies [3]. Compared with CT, the downside of a CXR is the overlapping anatomies in 2D projections. The high-contrast bony structures are one of the major interferences in CXRs, obscuring the underlying soft tissues. Therefore, the suppression of bone shadows in CXRs is highly desired to improve the conspicuity of lung tissues and, subsequently, improve diagnoses.

Recently, clinical evidence indicates that rib-removed CXRs can improve the diagnosis of various pulmonary abnormalities [4,5]. In a post-processing setup, previous approaches can be summarized as: (1) a dual-energy CXR that exposes the patient twice to a high and low tube voltage for digital subtraction of the ribs [6]; (2) unsupervised physical models [3,7]; and (3) deep learning (DL) models with supervision generated from dual-energy (DE) CXRs or a domain-adapted digitally reconstructed radiograph (DRR) [8,9].

A dual-energy CXR exploits the energy-dependent X-ray tissue interactions in the ribs and soft tissues and, then, performs digital subtraction to suppress the bone shadow. The method requires a specialized instrument that is not widely available and exposes the patients to a higher imaging dose, due to the dual exposure. Physical rib suppression models utilize a single-energy (SE) CXR first to reconstruct the rib structures by empirically assuming a rib pixel intensity distribution and then subtracting the rib shadows from the original CXR. For instance, von Berg et al. proposed the transfer of the ribs into the “ST-space”, where the contour of the rib appears as a straight line. Partial derivative computation, smoothing, and reintegration were then performed along the rib contours up to the centerline [7]. These highly hand-crafted rib suppression methods can preserve the morphological detail but are not scalable for clinical use, due to the tedious manual annotation of each rib and the case-by-case hyperparameter (HP) fine tuning that demands an impractically long processing time, which diminishes the benefit of CXRs [4,7,10,11,12].

DL methods are a time efficient alternative to conventional operations, and their performance depends on the methods and the training data quality [13]. Except for diverse network schemes for superior representation learning, the core disparities in existing DL approaches for CXR rib suppression are in the modality of the training images and training pair generation. Most of the previous models were trained on either DECXR or DRR. Specifically, since DECXR is formed by two XR exposures at distinct energy levels, it can later be decomposed into “soft-tissue-only” and “bone-only” images [14,15]. Learning models trained on DECXR form image pairs with unprocessed images as the input and decomposed “soft-tissue-only” images as the ground truth (GT) [3,16]. However, DECXR-based frameworks are limited by data scarcity [17]. Since the ribcage is separatable using computed tomography (CT), DRR-based methods attempt to project training pairs from the original and rib-suppressed CT [18]. Although DRRs simulate CXRs, their contrast and resolution differ substantially regarding the physical image formation details, detector signal processing, and post-processing [9]. The lack of paired images also hinders the domain adaptation training from a DRR to a CXR. Insufficiently trained DL approaches cannot learn the true mapping from raw to rib-removed CXRs. Consequently, the shadows of the rib edges are prevalent in DL predictions [3,9], adversely affecting the efficacy of downstream diagnostic tasks.

We contend that the answer to the rib suppression challenge lies in combining the two existing approaches. Inefficient physical models can maximally suppress the rib structures and preserve the lung tissue details, thus they are suitable to provide machine learning (ML) training input as a one-time investment. With such high-quality input, efficient DL models can be trained for wide clinical implementation.

We hereby introduce a benchmark dataset named FX-RRCXR for DL-based CXR rib suppression. FX-RRCXR is generated from the VinDr-RibCXR dataset [19], adapting the physical ST smoothing by von Berg et al. [7]. Next, we propose a supervised image denoising network, SADXNet, trained on FX-RRCXR for fast rib suppression in unseen CXRs. Lastly, we validate the impact of rib-removed CXRs in the detection of lung nodules using the NODE21 [20] dataset, and the classification and localization of benign lung disease using ChestX-ray14 [21] datasets. All the detection tasks are run on the Mask R-CNN framework [22].

## 2. Method

In this section, we will introduce the ST-smoothing method, FX-RRCXR dataset, SADXNet, and downstream validation of the NODE21 and ChestX-ray14 datasets.

### 2.1. ST Smoothing

The ST-smoothing algorithm assumes that the pixel intensities along one continuous contour of a rib are theoretically identical. As shown in Figure 1 S0, if the distances to the centerline of $p_{1}$ and $p_{2}$ are equal, these two points are deemed to be on the same contour and their corresponding rib intensities are equal.

#### 2.1.1. From Image Space to ST-Coordinate System

We explain Figure 1 S0→S1(a) here with Equations (1) and (2).
(1)$$T_{C}:\left(x,\;y\right)⟼\left(s,\;t\right)$$
(2)$$T_{C}^{-1}:\left(s,\;t\right)⟼\left(x,\;y\right)$$

ST transformation $T_{C}$ is a domain transformation used to generate a specific representation of a part of an image defined by the given closed cyclic contour $C:\;\gamma \left(t\right),\;t\in \left[0,\;C_{len}\right)$, $\gamma \left(0\right)=\gamma \left(C_{len}\right)$. $T_{C}$ is defined by its inverse as follows:(3)$$T_{C}^{-1}\left(s,\;t\right)=\gamma \left(t\right)+s\cdot {\frac{\gamma^{\prime }\left(t\right)}{\left|\gamma^{\prime }\left(t\right)\right|}}^{⊥}$$
where ${\frac{\gamma^{\prime }\left(t\right)}{\left|\gamma^{\prime }\left(t\right)\right|}}^{⊥}$ is the contour norm at γ(t). Figure 2a illustrates the transformation of contour C [7].

To reduce the computation burden, a continuous rib contour is considered as a piecewise linear contour (Figure 1 S0) for the discrete implementation of Equation (1). Figure 2b illustrates the implementation, with s and t formulated as:(4)$$s=\left|\left|\vec{QQ^{\prime }}\right|\right|$$
(5)$$t=t_{prev}+\left|\left|\vec{QP_{i}^{\prime }}\right|\right|\cdot \frac{\left|\left|P_{i}P_{i+1}\right|\right|}{\left|\left|\vec{P_{i}^{\prime }}\vec{P_{i+1}^{\prime }}\right|\right|}$$
where $t_{prev}$ is the sum of the length of all previous edges. The pixel intensities in the ST space are as follows:(6)$$I_{st_{C}}\left(s,t\right)=I(T_{C}^{-1}\left(s,\;t\right))$$
(7)$$\left(\hat{s},\hat{t}\right)=T_{C}(T_{C}^{-1}\left(s,t\right))$$

When it is assumed that a valid (s,t) always has a position $\left(\hat{s},\hat{t}\right)$ on the other side of the bone centerline c(t), we can obtain c(t) from:(8)$$c\left(t\right)=\underset{S}{\max}\forall \left(\hat{s},\hat{t}\right)\neq \left(s,t\right)$$

#### 2.1.2. Rib Extraction via Partial Derivatives Smoothing in ST Space

The current subsection explains steps S1(a)→S2 in Figure 1.

##### Discrete Partial Derivative in ST Space

As shown in Equation (9), the first-order partial derivative is calculated along the s axis in a discrete form to boost the overall computation. The definition of $I_{d_{C}}\left(s,\;t\right)$ represents the gradient orthogonal to the t axis of a rib, which means that any structure oriented along axis t does not contribute to the bone gradients.
(9)$$I_{d_{C}}\left(s,\;t\right)=\partial sI_{st_{C}}\left(s,t\right)=I\left(s,t\right)-I\left(s-1,\;t\right)$$

##### Smoothing, Reintegration, and Transformation Back to the XY Domain

Improved from von Berg et al. [7], Gaussian smoothing ($G_{{𝓀}_{t}}$) along the t axis at $I_{d_{C}}$ and centerline smoothing (C) along the s axis at the reintegrated $I_{r_{C}}$ are implemented. First, since we hypothesize that signals along the t axis of $I_{d_{C}}$ are independent from the ribs. A large Gaussian kernel ${𝓀}_{t}\;$ was used to smooth out the signals from the soft tissue and leave only the signals from the ribs in $G_{{𝓀}_{t}}\left(I_{d_{C}}\right)$. Note that ${𝓀}_{t}$ is a HP. Next, after excluding the soft tissue signal via $G_{{𝓀}_{t}}\left(I_{d_{C}}\right)$, we reintegrate towards the smoothed partial gradients to recover the bone signals $I_{rC}\left(s,t\right)$ in the ST domain, as shown in Equation (10).
(10)$$I_{r_{C}}\left(s,\;t\right)=\int_{s-1}^{s}G_{{𝓀}_{t}}\left(I_{d_{C}}\left(s,t\right)\right)+G_{{𝓀}_{t}}\left(I_{d_{C}}\left(s-1,t\right)\right)$$

Lastly, because $T_{C}$ has a singularity at the centerline c(t), an artificial edge was observed along c(t) , as shown in Figure 1 S2, after the reintegration of $I_{r_{C}}$. Therefore, a K-nearest neighbor (KNN) based centerline smoothing is applied along the s axis of $I_{r_{C}}$ to smooth out the artificial edge, according to Equation (11).
(11)$$C\left(I_{r_{C}}\left(s_{i},t_{j}\right)\right)=\{\begin{array}{c} I_{r_{C}}\left(s_{i},t_{j}\right),\;\;if\;\frac{I_{r_{C}}\left(s_{i},t_{j}\right)}{I_{r_{C}}\left(s_{i-1},t_{j}\right)}>\tau \\ \frac{\sum_{m=i-k+1}^{i}I_{r_{C}}\left(s_{m},t_{j}\right)}{k},\;otherwise \end{array}$$
where τ and k are two HPs and represent the threshold for conducting the KNN average and the number of neighbors used, respectively.

After the above steps, we transfer the rib intensity $C\left(I_{r_{C}}\right)$ from the ST back into the image space under Equation (12) to exclude possible negative values, which are uninterpretable in the image space.
(12)$$I_{bone_{C}}=\max\left(C\left(I_{r_{C}}\right),\;0\right)$$

#### 2.1.3. Rib Removal and Border Blending

We focus on Figure 1 S3 here. The initial rib-suppressed CXR $I_{soft_{C}}$ is acquired by subtracting the $I_{bone_{C}}$ from the raw CXR, as follows:(13)$$I_{soft_{C}}\left(x,y\right)=I\left(x,y\right)-I_{bone_{C}}\left(x,y\right)$$

To improve the continuity between the rib boundary $r_{b}$ and its surrounding soft tissues, a KNN border smoothing function is applied to $r_{b}$:(14)$$r_{b}^{\prime }\left(x,y\right)=\frac{\sum_{i=1}^{k}r_{b,i}\left(x,y\right)}{k}$$
where $r_{b}$ is defined under the ST space and shown in Equation (15). $s_{b}$ in Equation (15) and k in Equation (14) are two HPs.
(15)$$I_{soft_{C}}\left(s,t\right)\;in\;r_{b}\left(s,t\right),\;\;if\;s\leq s_{b}\;r_{b}\left(x,y\right)=T_{C}^{-1}\left(r_{b}\left(s,t\right)\right)\;$$

Lastly, generating a complete soft tissue CXR requires iteratively repeating Section 2.1.1, Section 2.1.2 and Section 2.1.3 for each rib in a complete ribcage.

### 2.2. Data Cohort

#### 2.2.1. VinDr-RibCXR Dataset

VinDr-RibCXR is selected for creating the FX-RRCXR dataset using our modified ST-smoothing algorithm. VinDr-RibCXR is a dataset for automatic rib segmentation and labeling of SECXR scans. It contains 245 images with corresponding rib masks annotated by an expert. Each CXR scan has 20 separate rib annotations in the left and right lungs, respectively. The dataset was pre-split into training and validation sets, with 196 scans in the training set and 49 in the validation set. We refer the readers to Nguyen et al. [19] for more details.

#### 2.2.2. FX-RRCXR Dataset Preparation

We applied the ST-smoothing algorithm to 245 images from the VinDr-RibCXR dataset. The HPs required by ST smoothing are tuned for individual images using the random grid search algorithm. Excluding the process for HP searching, the ST smoothing took 40–70 min to process a rib-removed scan, depending on the number of pixels within the ribcage. We organized the original CXRs as input and their corresponding rib-suppressed scans as GT while preparing the image pairs and kept the same training and validation split as the VinDr-RibCXR.

### 2.3. SADXNet

Since rib signals can be treated as noise superimposed on the rib-suppressed CXRs, we proposed the design of an image-denoising network, named SADXNet, to be trained on the FX-RRCXR dataset. Inspired by the architecture of DenseNet [23], SADXNet is designed to be densely connected, as shown in Figure 3. For each layer in SADXNet, the feature maps of all the preceding layers and their own feature map are fed into all the subsequent layers. The advantage of dense connection is the better alleviation of the vanishing-gradient problem, strengthening feature propagation, and encouraging feature reuse [23]. The composition of SADXNet is detailed below.

**Dense connectivity**. Figure 3 shows the layout of the densely connected SADXNet schematically. Each layer can be described by Equation (16):(16)$$x_{l}=N_{l}\left(\left[f_{c_{l}}\left(x_{0}\right),f_{c_{l}}\left(x_{1}\right),\ldots ,f_{c_{l}}\left(x_{l-2}\right),x_{l-1}\right]\right)$$
where $\left[x_{0},x_{1},\ldots ,x_{l-1}\right]$ represents the channel-wise concatenation of the feature maps produced from layer 0 to l−1 and $f_{c_{l}}\left(\cdot \right)$ is a 1×1 convolution (Conv) to unify the number of feature channels of $x_{0,}\ldots ,$ $x_{l-2}$. The output channel of $f_{c_{l}}$ is defined by $\frac{C\left(x_{l-1}\right)}{L-1}$, where C represents the number of feature channels of $x_{l-1}$. The purpose of $f_{c_{l}}\left(\cdot \right)$ is to avoid the concatenated input feature maps, $N_{l}\left(\cdot \right)$ being overly large in channel dimension exceeding the GPU memory.

**Composite function:** We define $N_{l}\left(\cdot \right)$ as a composite function of three consecutive operations: batch normalization (BN), rectified linear unit (ReLU), and 3 × 3 convolution (Conv).

**Pooling layers:** Inspired by a pioneering study [8], we preserve the height (H) and width (W) dimensions in the feature maps as the shape of the input images throughout the Conv process without using down- or up-sampling layers.

**Channel design:** Overall, there are seven densely connected layers in the SADXNet. The channel of the convolutional kernel for each layer is designed in an increase-to-decrease setting to mimic the design of a fully convolutional network [24] in regard to channel dimension, so as to strike a balance between the model complexity (kernel with more channels) and training time (kernel with fewer channels).

**Loss function:** The cost function of the SADXNet is designed as a combination of the negative peak signal-to-noise ratio, the multi-scale structure similarity index measure [25], and the $L_{1}$ deviation measurement:(17)$$L=-\alpha \cdot L_{PSNR}+\left(1-\alpha \right)\cdot \left[\beta \cdot L_{MS-SSIM}+\left(1-\beta \right)\cdot L_{1}\right]$$
(18)LPSNR=log10(MAXX21mn·∑i=0m−1∑i=0n−1[xij−yij]2)
(19)$$L_{MS-SSIM}=1-\frac{\left(2\mu_{x}\mu_{y}+c_{1}\right)}{\left(\mu_{x}^{2}+\mu_{y}^{2}+c_{1}\right)}\cdot \prod_{j=1}^{M}\frac{\left(2\sigma_{x_{j}y_{j}}+c_{2}\right)}{(\sigma_{x_{j}}^{2}+\sigma_{y_{j}}^{2}+c_{2)}}$$
(20)$$L_{1}=\frac{1}{mn}\cdot \sum_{i=0}^{m-1}\sum_{i=0}^{n-1}{\left|\left|x_{ij}-y_{ij}\right|\right|}_{1}\;$$
where α and β in Equation (17) are HPs set to 0.75 and 0.25, X and Y are the model input and target, $MAX_{X}$ is the maximum possible input value, $\left[\sigma_{x_{j}y_{j}},\;\ldots ,\sigma_{x_{M}y_{M}}\right]$ of Equation (18) set to [0.5, 1.0, 2.0, 4.0, 8.0], $c_{1}={\left(k_{1}S\right)}^{2}$ and $c_{2}={\left(k_{2}S\right)}^{2}$ of Equation (19) are two variables to stabilize the division with a weak denominator having S as the dynamic range of the pixel values (typically $2^{\#\;bits\;per\;pixel}-1$) and ($k_{1}$, $k_{2})$ as the constants, and ${\left|\left|\cdot \right|\right|}_{1}$ denotes the $l_{1}$ norm.

**Model training:** SADXNet was implemented in PyTorch, and the training was performed on a GPU cluster with 4×RTX A6000. Per the SADXNet training, we set the maximum epoch number to 200 and observed that the model converged at around 100 epochs. The Adam optimizer with an initial learning rate (LR) of 0.001 and a batch size of 1 × 4 were applied.

**Evaluation metrics**: The root mean square error (RMSE) is used for evaluating the SADXNet performance with the corresponding GT generated from ST Smoothing.

### 2.4. Downstream Clinical Task Validation

We quantified the benefits of rib-suppressed images with two experiments, including lung nodule detection and a general pulmonary disease classification and localization task based on the NODE21 [20] and ChestXRay14 [21] datasets, respectively. The details are set out below.

**Training input**: We organized the input into three combinations: (1) solely raw CXRs; (2) solely soft-tissue CXRs; and (3) mixed raw and rib-suppressed CXRs. All the rib-suppressed images were predicted using SADXNet.

**Datasets**: NODE21 [20] encompasses 4882 CXR scans with the ratio of patient:volunteer = 1134:3748. Moreover, 5524 annotations were made to these images, with a maximum of three positive annotations for each scan. ChestX-ray14 [21] is a CXR set that has been text mined with fourteen lung diseases and bounding box annotations for 984 images in the pre-split test set. We extracted the 984 annotated images and then randomly sampled 3×984 healthy volunteers from the test set to construct a dataset for supervised detection training. For both datasets, we split the training and validation sets in the ratio of 7:3 and carefully balanced the proportion of positive and negative cases.

**Detection network**: Both tasks were performed in the Mask R-CNN pipeline and implemented in PyTorch [22], training the network separately on three different types of input with data augmentation of random scaling, random cropping, and random Gaussian blur, an LR of 0.001, a stochastic gradient descent optimizer, and a batch size of 2 × 4 on the 4×RTX A6000 GPU cluster. Altogether, we ran 30,000 and 50,000 training iterations for NODE21 [20] and ChestX-ray14 [21], respectively.

**Evaluation metrics**: The area under the curve (AUC), the true positive predictions (TP), the false positive predictions (FP), and the false negative predictions (FN) are used as evaluation metrics.

## 3. Results

### 3.1. Rib Suppression

#### 3.1.1. ST Smoothing

Figure 4 demonstrates a patient with lung nodules in the left lung and compares the node visibility with and without rib removal using ST smoothing. Compared to previous rib-suppression methods with edge residuals or artifacts near the lung borders [4,9], ST smoothing carefully avoids those two drawbacks. Additionally, ST smoothing preserved the shape and morphological details of the lung tumors.

#### 3.1.2. SADXNet

Figure 5 shows two subjects predicted by SADXNet. We found that the rib-suppressed scans predicted by SADXNet are visually indistinguishable from their corresponding GTs. Quantitatively, SADXNet achieves a $2.32\pm 0.13\times 10^{-5}$ test RMSE. Most notably, compared to the time consuming ST-smoothing algorithm, SADXNet suppresses one scan in <1 s.

### 3.2. Downstream Clinical Task Validation

#### 3.2.1. NOD21 Detection

According to Table 1, mixing raw and rib-suppressed images in training achieved the best detection scores than training on only a single source input. We also evaluated the mix-trained model on raw and rib-suppressed validation sets, respectively. Both scenarios achieved comparable performance, with slightly better outcomes for the raw CXR. Quantitatively, training using mixed images achieved approximately a 2–3% higher AUC, located more nodules (Figure 6 first row), and significantly reduced the FP (Figure 6 second row) than single-source images. Lastly, the performance of networks trained on a single image type is similar, with a slightly lower FP using rib-suppressed images.

#### 3.2.2. ChestX-ray14 Classification and Localization

As shown in Table 2, training with mixed scans achieved the best performance across the three input combinations, resulting in around a 6–7% higher AUC than single-source trained detectors and largely reducing the FP predictions. Single-source trained models roughly reach similar performance, except that a model trained with rib-suppressed images makes fewer FP classifications. Figure 6 visually confirms the quantitative results.

Moreover, since ChestX-ray14 is for multiple lung disease localization, we also present the AUC for each disease in Table 3. Despite the varying disease-specific performance, the improvements benefiting from mixed training are consistent with Table 2.

## 4. Discussion

Chest X-rays (CXRs) are highly accessible and cost effective for point-of-care pulmonary disease screening. However, the interpretation of a CXR can be difficult because structures overlap with each other in 2D projection. Specifically, tissue details can be obscured by bright ribs in a CXR due to their higher atomic number and density. We hereby combined physical and ML methods’ strengths to overcome their weaknesses. We first used ST smoothing and the VinDr-RibCXR [19] dataset to create a benchmark dataset, FX-RRCXR, with 245 paired original and rib-suppressed CXRs. We then trained a supervised denoising network, SADXNet, achieving high-quality sub-second rib suppression. Lastly, we evaluated the quality of the FX-RRCXR dataset and SADXNet using two downstream tasks, including lung nodule detections using the NODE21 [20] dataset and fourteen lung disease classifications and localization using the ChestXray14 [21] dataset.

Our contribution is thus twofold. First, we used a physical model to generate a qualitative dataset that supports deep learning, which is released to the public as part of the paper. Second, we trained a fully automated supervised deep network that achieved physical model quality 1 × 10^4^ times faster. The enormous gain in efficiency benefits large-scale testing of rib-suppressed CXR for various clinical applications, two of which were exemplified here, including the testing efficacy of lung nodule detection and benign disease classification. In both end-to-end tests, our quantitative testing showed significant improvement in diagnosis with rib suppression. To the best of our knowledge, this is the first study to demonstrate the benefit of combining a physical method with ML in end-to-end diagnostic testing.

The current study also demonstrated the pipeline’s robustness. Generally, distributional shift poses a major challenge in clinical deployment [26]. Here, we found that although SADXNet was trained on FX-RRCXR, it still robustly suppressed the rib structures on scans from NODE21 and ChestX-ray14 (see Figure 6 middle column), yielding improved stability in clinically relevant tasks.

An interesting observation is that training a detection model on rib-suppressed CXRs alone does not improve model performance compared to training on original scans. In contrast, mixed training of two image sources can significantly reduce type I error while moderately increasing sensitivity. We attribute this fact to three considerations: (1) mixed training achieved better performance, namely a comparison in the region of interest (ROI) with and without the rib helped the learning model; (2) there was a significant decrease in type I error, namely assistance from the rib-removed scans makes the model more likely to avoid misidentifying noise in the rib structures, such as edges, as an ROI; and (3) there was a moderate reduction in type II error, namely the ribs only account for part of the superimposing anatomy in CXRs. Rib Suppression does not reduce interference from other anatomical and non-anatomical structures, including the heart, major vessels, mediastinum, and attached sensors. However, the current study may provide a roadmap for reducing additional interference for CXR-based diagnostic tasks.

## 5. Conclusions

We improved ST smoothing from von Berg et al. [7] and, based on that, further introduced a paired dataset, FX-RRCXR, that serves as a benchmark for supervised DL on CXR rib removal. Next, we proposed a denoising network, SADXNet, which learnt rib suppression by considering the ribs as noise on the lung tissues. Lastly, we validated the efficacy of rib suppression using two downstream tasks, including lung nodule detection and common lung anomaly classification and localization. The experimental results from the downstream tasks quantitatively substantiated the benefit of the FX-RRCXR dataset and SADXNet.

## Figures and Tables

**Figure 1 diagnostics-13-01652-f001:**
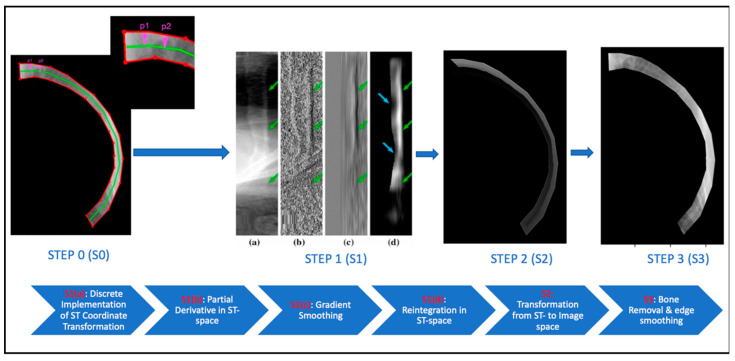
Major steps in the ST-smoothing algorithm applied to the fifth rib in the CXR with image visualization in the upper and pipeline description in the bottom section. The explanation for the purpose of each image is labeled in red on the flow chart.

**Figure 2 diagnostics-13-01652-f002:**
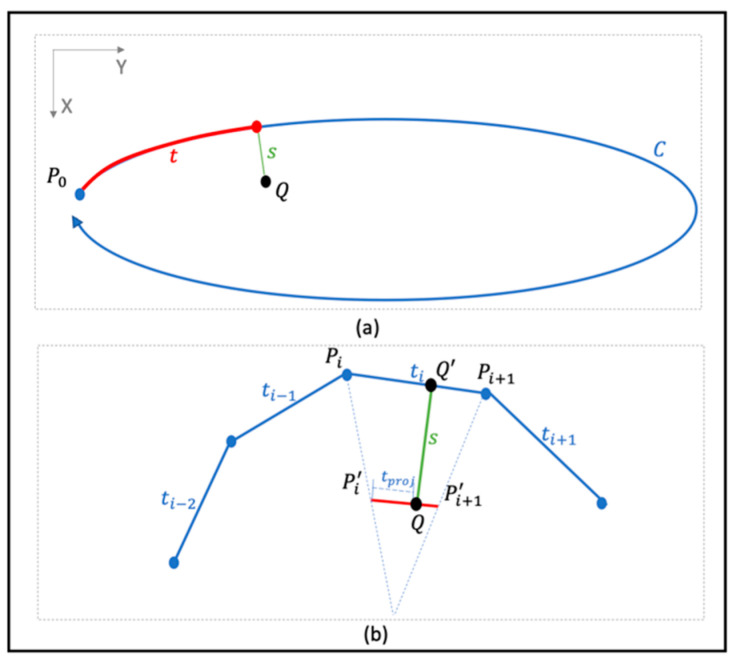
Visualization of the ST-coordinate system. (**a**) Point Q transformation from XY- into ST domain regarding contour C. (**b**) Discrete sampling in ST-coordinate system with respect to point Q.

**Figure 3 diagnostics-13-01652-f003:**
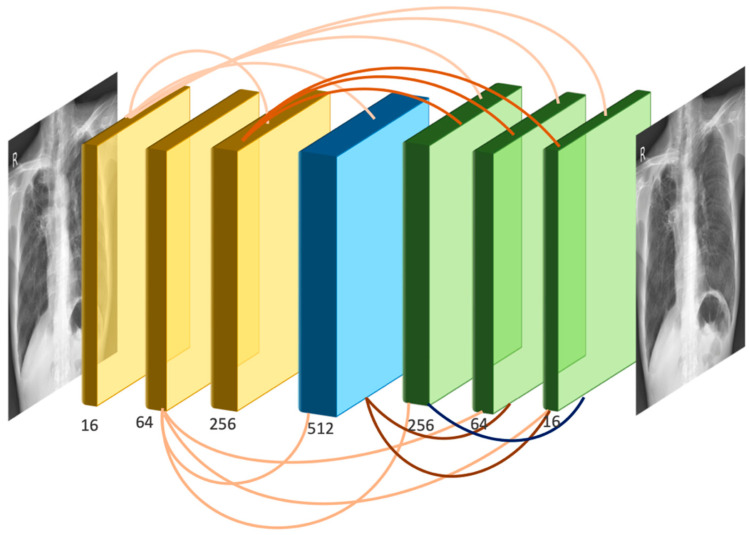
The network architecture of SADXNet.

**Figure 4 diagnostics-13-01652-f004:**
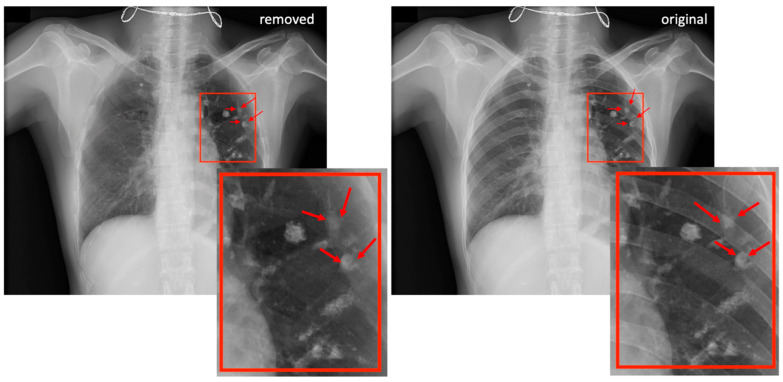
A sample case involving ST smoothing with several lung nodules in the left lung field. The left-hand side shows the rib-removed scan, and the right-hand side shows the original unprocessed CXR image. Red arrows denote radiologically confirmed nodules.

**Figure 5 diagnostics-13-01652-f005:**
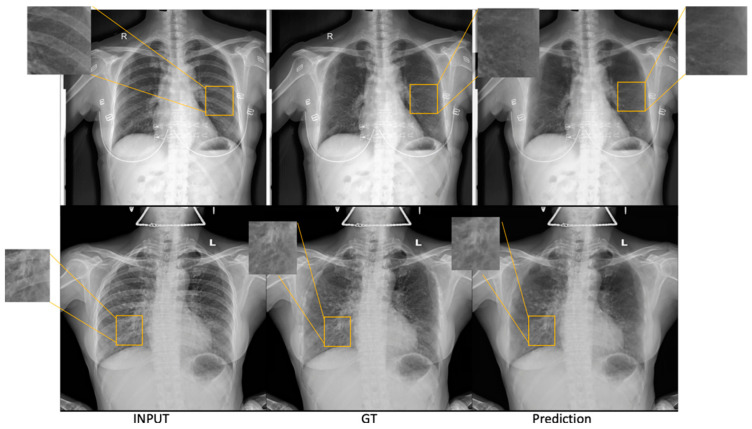
Two sample cases from the test set illustrate the results of SADXNet rib removal. Each row represents one patient. The columns from left to right are ordered as the first column showing the raw CXR, the second showing the rib-suppressed image generated from ST smoothing (GT for SADXNet training supervision), and the third showing the rib-suppressed CXRs predicted by the trained SADXNet model.

**Figure 6 diagnostics-13-01652-f006:**
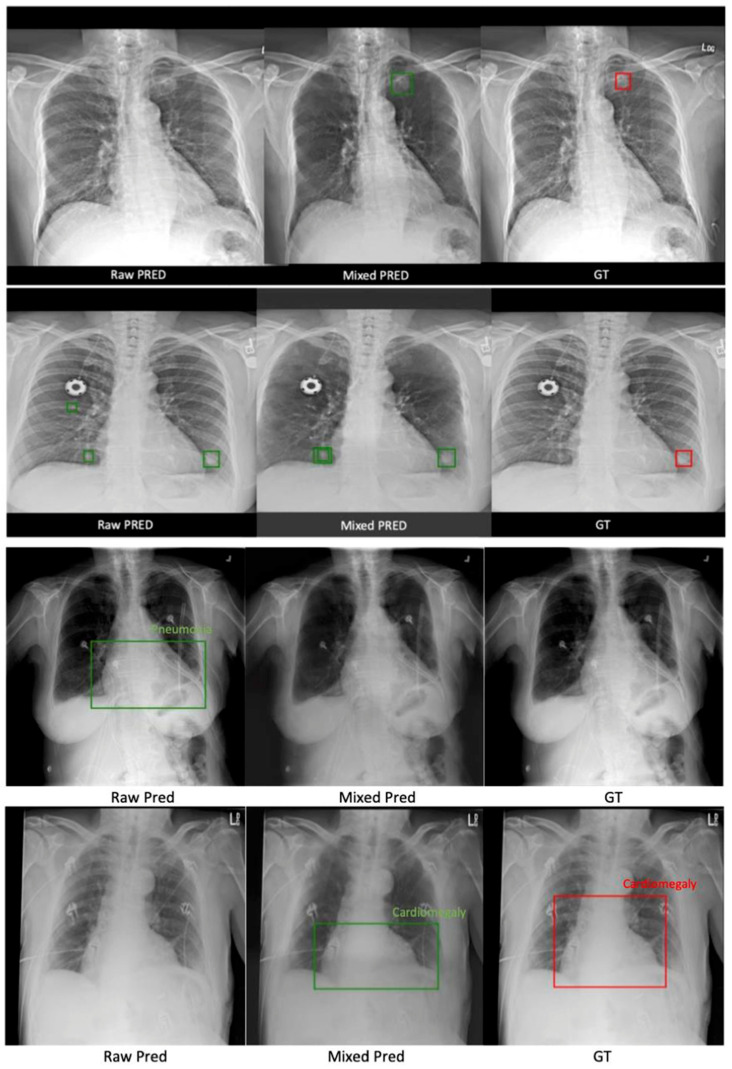
Two sample cases from the NODE21 test set in the first two rows, and two sample cases from the ChestX-ray14 test set in the third and fourth rows. The figure is organized into three columns: the first column shows predictions made using the model trained with raw CXRs only, the second column shows predictions made using the mix-trained model, and the last column shows the GTs. Red boxes denote manually labeled bounding boxes for the disease location. Green boxes are results of automated detection.

**Table 1 diagnostics-13-01652-t001:** Evaluation results from the NODE21 dataset. FN, FP, and TP show the absolute number of predictions for a more straightforward comparison. The best performer is marked in bold.

Modality	Training Input	Validation Input	AUC	FN	FP	TP
Mask R-CNN	Raw	Raw	94.76%	48	1273	372
	Rib suppressed	Rib suppressed	95.32%	45	1193	375
	Raw + Rib suppressed	Raw	**97.99%**	**32**	**1070**	**388**
		Rib suppressed	97.31%	33	1082	387

**Table 2 diagnostics-13-01652-t002:** Evaluation results from the ChestX-ray14 dataset. FN, FP, and TP show the absolute number of predictions for a more straightforward comparison. The best performer is marked in bold, and the lowest result is underlined.

Modality	Training Input	Validation Input	AUC	FN	FP	TP
Mask R-CNN	Raw	Raw	80.54%	137	3029	245
	Rib suppressed	Rib suppressed	81.55%	138	2909	244
	Rib suppressed	Raw	**87.16%**	**116**	**2644**	**275**
		Rib suppressed	86.89%	124	2701	267

**Table 3 diagnostics-13-01652-t003:** The AUC scores for 14 lung diseases in the ChestX-ray14 dataset. Tr/Val represents training/validation. The best results are marked in bold, and the lowest ones are underlined.

	Input: Tr/Val	Raw/Raw	ST/ST	Raw + ST/Raw	Raw + ST/ST
Disease					
Atelectasis		79.21%	77.35%	**85.27%**	84.89%
Cardiomegaly		83.47%	82.03%	**93.84%**	92.67%
Effusion		84.98%	84.72%	**91.02%**	**91.02%**
Infiltration		69.84%	70.00%	**75.21%**	74.32%
Mass		77.23%	77.38%	**87.83%**	85.79%
Nodule		76.32%	75.37%	**82.33%**	81.29%
Pneumonia		73.89%	73.24%	**79.43%**	78.33%
Pneumothorax		82.43%	82.32%	**91.37%**	90.62%
Consolidation		74.57%	75.23%	**82.77%**	81.98%
Edema		88.14%	87.28%	95.32%	**95.33%**
Emphysema		90.68%	89.47%	**96.21%**	95.79%
Fibrosis		82.99%	83.21%	**87.45%**	86.76%
Pleural Thickening		73.37%	72.45%	**80.34%**	79.85%
Hernia		88.03%	89.21%	**97.24%**	96.79%

## Data Availability

The data will be available at https://github.com/FluteXu/CXR-Rib-Suppression (accessed on 9 April 2023).

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
