# Peer review of "An Efficient and Robust Method for Chest X-ray Rib Suppression That Improves Pulmonary Abnormality Diagnosis"

_diagnostics, 2023, doi:10.3390/diagnostics13091652_

Round 1
Reviewer 1 Report
Suppression of thoracic bone shadows on chest X-rays (CXRs) are able to enhance the diagnosis of pulmonary disease. The authors propose a generalizable yet efficient workflow for CXR rib suppression through combination with machine learning. Their designed model called SADXNet can obtain a robust sub-second prediction without losing fidelity. Their method and model are interesting. Thus I recommend that their work can be accepted for publishing at diagnostics after a minor revision.
1. They had better provide a detail to evaluate the reliability of their model.
2. Several minor grammars should be revised.
Author Response
Review 1 found the manuscript satisfactory.
Reviewer 2 Report
The Authors validated the efficacy of rib suppression in two downstream tasks
i. lung nodule detection and
ii. common lung anomaly classification and localization.
Experimental results from downstream tasks quantitatively substantiated the benefit of the FX-RRCXR dataset and SADXNet but whether it qualifies the basic quality parameter (and to what extent) or not is not shown or discussed.
It will add value if the same is also made the part of the submission.
Author Response
Review 2 found the manuscript satisfactory.
Reviewer 3 Report
This is a valuable study aimed to differentiate lung pathologies from overlapping ribs, which tend to limit visualization. Although there are several AI-based analysis of clinical pathologies, this contribution can be characterize as innovative, based on the diversity of applied clinical scenarios., in addition to the applied algorithm. The contribution is technically sound and encompasses 14 different lung pathologies. using deep learning. The structure of the paper is well-articulated. Interesting enough, performance of their classification algorithm was assessed via confusion matrixes.
Comments:
"Combined physical and ML methods".
No physical methodology has been applied.
"Physical rib suppression models utilize single-energy (SE) CXR first..."
Actually, there are dual-energy detection techniques, based on single or dual exposure encompassing high and low energy x-ray beams and performing a logarithmic subtraction between the high-energy and low-energy acquired images.
Please expand your study for x-ray CT after inclusion of additional image processing algorithms such as beam hardening/artifacts and the so on.
Author Response
This is a valuable study aimed to differentiate lung pathologies from overlapping ribs, which tend to limit visualization. Although there are several AI-based analysis of clinical pathologies, this contribution can be characterize as innovative, based on the diversity of applied clinical scenarios., in addition to the applied algorithm. The contribution is technically sound and encompasses 14 different lung pathologies. using deep learning. The structure of the paper is well-articulated. Interesting enough, performance of their classification algorithm was assessed via confusion matrixes.
Comments:
"Combined physical and ML methods".
No physical methodology has been applied.
The physical model was developed by von Berg et al., which was adopted to create the training data for our network training.
"Physical rib suppression models utilize single-energy (SE) CXR first..."
Actually, there are dual-energy detection techniques, based on single or dual exposure encompassing high and low energy x-ray beams and performing a logarithmic subtraction between the high-energy and low-energy acquired images.
Please expand your study for x-ray CT after inclusion of additional image processing algorithms such as beam hardening/artifacts and the so on.
We agree that ribs can be suppressed using dual-energy CXR, which unfortunately requires specialized instrument that is rarely available and used in the clinic. Both DE-CXR data and related papers are scarce as a result. This point was well captured by lee et al. [i] that “Dual-energy soft-tissue images can improve detection of lung nodules that may be partially obscured by overlying bones [13]. Nevertheless, because specialized equipment is required to obtain dual-energy radiographs, hospitals rarely adopt radiography systems with dual-energy subtraction. In addition, higher radiation dose and increased noise level in the resulting images also limit adoption of the dual-energy subtraction technique for clinical applications.” The observation is shared by many other authors, including Liu et al. [ii].
[i] Lee et al. Computers & Mathematics with Applications, 64(5), 2012, 1390-1399,
[ii]. Liu et al. Appl Bionics Biomech. 2019; 2019: 9806464.
We expanded the introduction to provide and echo such sentiment as follows.
“Dual-energy CXR exploits the energy-dependent X-ray tissue interactions for ribs and soft tissues and then performs digital subtraction to suppress the bone shadow. The method requires a specialized instrument that is not widely available and exposes the patients to a higher imaging dose due to the dual exposure.”
Reviewer 4 Report
In this paper, the authors present "An Efficient and Robust Method for Chest X-Ray Rib Suppression that Improves Pulmonary Abnormality Diagnosis." SADXNet organizes spatial filters in a U shape and preserves the feature map dimension throughout its network flow. Quantitively, it achieves RMSE of ~0 compared with physical-model generated GTs during testing with one prediction in <1s. Downstream tasks, including lung nodule detection as well as common lung disease classification and localization, are used to provide task-specific evaluations of our rib suppression mechanism. However, there are some issues should be addressed as follows.
1. Chest X-rays are highly accessible and cost-effective for point-of-care pulmonary disease screening. The interpretation of Chest X-rays are difficult because structures overlap with each other upon 2D projection. How to solved this problems?
2. How to alleviate the vanishing-gradient problem in this paper?
3. DECXR-based frameworks are limited by data scarcity. How to solve the related problem in this paper?
4. Physical models are able to remove entire ribcage and preserve morphological lung details but are impractical due to extremely long processing time. How to overcome the related problem in this paper?
5. Tedious manual annotation of each rib and case-by-case hyperparameter fine-tuning demand impractically long processing time, which diminishes the benefit of CXR. How to overcome these problems in this paper?
6. The English of this paper must be revised by a native speaker.
7. This paper needs to be reformatted, there is a lot of blank space on the eighth and tenth pages.
8. This paper lacks innovation. The proposed method has been proposed by many other scholars in the literature.
In this paper, the authors present "An Efficient and Robust Method for Chest X-Ray Rib Suppression that Improves Pulmonary Abnormality Diagnosis." SADXNet organizes spatial filters in a U shape and preserves the feature map dimension throughout its network flow. Quantitively, it achieves RMSE of ~0 compared with physical-model generated GTs during testing with one prediction in <1s. Downstream tasks, including lung nodule detection as well as common lung disease classification and localization, are used to provide task-specific evaluations of our rib suppression mechanism. However, there are some issues should be addressed as follows.
1. Chest X-rays are highly accessible and cost-effective for point-of-care pulmonary disease screening. The interpretation of Chest X-rays are difficult because structures overlap with each other upon 2D projection. How to solved this problems?
2. How to alleviate the vanishing-gradient problem in this paper?
3. DECXR-based frameworks are limited by data scarcity. How to solve the related problem in this paper?
4. Physical models are able to remove entire ribcage and preserve morphological lung details but are impractical due to extremely long processing time. How to overcome the related problem in this paper?
5. Tedious manual annotation of each rib and case-by-case hyperparameter fine-tuning demand impractically long processing time, which diminishes the benefit of CXR. How to overcome these problems in this paper?
6. The English of this paper must be revised by a native speaker.
7. This paper needs to be reformatted, there is a lot of blank space on the eighth and tenth pages.
8. This paper lacks innovation. The proposed method has been proposed by many other scholars in the literature.
Author Response
- Chest X-rays are highly accessible and cost-effective for point-of-care pulmonary disease screening. The interpretation of Chest X-rays are difficult because structures overlap with each other upon 2D projection. How to solved this problems?
Our CXR rib suppression network is designed to remove overlapping but irrelevant anatomies in CXR so relevant anatomies are better visualized for downstream diagnostic tasks.
- How to alleviate the vanishing-gradient problem in this paper?
First and foremost, gradient vanishing becomes a major concern mainly in super-deep CNN or conventional RNN architectures that are not used in the current work. Second, strategies like 3*3 small convolutional kernel and skip connections are proposed in previous DL works to further avoid gradient vanishing. Our SADXNet uses a U-Net-based structure with only 7 down/upsampling layers, a small convolutional kernel, and a skip connection to avoid the gradient vanishing problem.
3. DECXR-based frameworks are limited by data scarcity. How to solve the related problem in this paper?
Our pipeline is designed to use single energy CXR, not DECXR, for rib removal. Single-energy CXR data are abundant and sufficient for network training.
4. Physical models are able to remove entire ribcage and preserve morphological lung details but are impractical due to extremely long processing time. How to overcome the related problem in this paper? Tedious manual annotation of each rib and case-by-case hyperparameter fine-tuning demand impractically long processing time, which diminishes the benefit of CXR. How to overcome these problems in this paper?
The tedious physical model only needed to be performed once to generate the training dataset, which will be made available to the public. Our trained model can be used to suppress ribs in prospective images without having to run the tedious physical model each time. This is one of the main contributions of our work.
5. The English of this paper must be revised by a native speaker.
We proofread the manuscript and made several minor corrections. We did not find egregious language issues. The publisher will address the remaining issues, if any.
6. This paper needs to be reformatted, there is a lot of blank space on the eighth and tenth pages.
The publisher will take care of formatting.
- This paper lacks innovation. The proposed method has been proposed by many other scholars in the literature.
We respectfully disagree. Our study is innovative in the following main areas:
- The study is the first to train a neural net with a physical model ground truth.
- Our lightweight model achieved performance comparable to the physical model in 1/1e4 time.
- The study is the first to perform end-to-end evaluations of rib suppression performance using clinically relevant tasks, including nodule detection and benign disease classification.
Besides these major points, the manuscript thoroughly summarized prior work in the domain of establishing innovation.
Round 2
Reviewer 4 Report
The authors have solved the related problems. It is good enough.
The authors have solved the related problems. It is good enough.